# Cardioprotection and Suppression of Fibrosis by Diverse Cancer and Non-Cancer Cell Lines in a Murine Model of Duchenne Muscular Dystrophy

**DOI:** 10.3390/ijms25084273

**Published:** 2024-04-12

**Authors:** Laris Achlaug, Irina Langier Goncalves, Ami Aronheim

**Affiliations:** Department of Cell Biology and Cancer Science, Ruth and Bruce Rappaport Faculty of Medicine, Technion—Israel Institute of Technology, P.O. Box 9649, Haifa 31096, Israel; laris1010.ab@gmail.com (L.A.); irinalan@campus.technion.ac.il (I.L.G.)

**Keywords:** Duchenne muscular dystrophy, fibrosis, cardiac remodeling, cardiac dysfunction

## Abstract

The dynamic relationship between heart failure and cancer poses a dual challenge. While cardiac remodeling can promote cancer growth and metastasis, tumor development can ameliorate cardiac dysfunction and suppress fibrosis. However, the precise mechanism through which cancer influences the heart and fibrosis is yet to be uncovered. To further explore the interaction between heart failure and cancer, we used the MDX mouse model, which suffers from cardiac fibrosis and cardiac dysfunction. A previous study from our lab demonstrated that tumor growth improves cardiac dysfunction and dampens fibrosis in the heart and diaphragm muscles of MDX mice. We used breast Polyoma middle T (PyMT) and Lewis lung carcinoma (LLC) cancer cell lines that developed into large tumors. To explore whether the aggressiveness of the cancer cell line is crucial for the beneficial phenotype, we employed a PyMT breast cancer cell line lacking integrin β1, representing a less aggressive cell line compared to the original PyMT cells. In addition, we examined immortalized and primary MEF cells. The injection of integrin β1 KO PyMT cancer cells and Mouse Embryo Fibroblasts cells (MEF) resulted in the improvement of cardiac function and decreased fibrosis in the heart, diaphragm, and skeletal muscles of MDX mice. Collectively, our data demonstrate that the cancer line aggressiveness as well as primary MEF cells are sufficient to impose the beneficial phenotype. These discoveries present potential novel clinical therapeutic approaches with beneficial outcome for patients with fibrotic diseases and cardiac dysfunction that do not require tumor growth.

## 1. Introduction

Fibrotic diseases contribute to approximately half of the mortality rate in developed countries and present an unresolved clinical need [1]. In recent studies, we demonstrated that injecting two different murine cancer cell lines representing breast and lung cancer resulted in the amelioration of cardiac dysfunction, reduced cardiac hypertrophy, and lower cardiac fibrosis, in a pressure overload [2] and transgenic mouse model for heart hypertrophy [3]. To further establish the suppression of fibrosis in other organs as well, we used the MDX mouse model for Duchenne muscular dystrophy (DMD). These mice suffer from cardiac dysfunction and systemic fibrosis in the skeletal, cardiac, and diaphragm muscles [4]. Cancer cell line implantation led to decreased fibrosis in the skeletal, heart, and diaphragm muscles of MDX mice, thereby leading to improved cardiac function [5]. This phenomenon was partially mediated by altered gene expression in the heart and other organs [2,5]. These findings were established using aggressive cell lines: Polyoma Middle T (PyMT) and Lewis lung carcinoma (LLC). PyMT murine breast carcinoma cells were generated in a transgenic murine model for metastatic breast cancer [6], while the LLC is a known cell line and widely used as a model for metastasis [7]. To reveal the role of cancer cell line aggressiveness for the beneficial phenotype, we utilized three different cell lines varying in their malignancy: PyMT Integrin β1 knockout (PyMT ITGB1 KO) and immortalized and primary Mouse Embryonic Fibroblasts (designated thereafter: MEF^I^ and MEF^P^, respectively). Integrin β1 is a key member of the integrin receptor family [8], widely expressed across cell types [9]. Integrins are crucial for tumor cell invasiveness and metastasis, as well as for regulating programmed cell death, influencing tumor growth and progression [10]. Deletion of *integrin β1* in human breast cancer cell lines has demonstrated reduced xenograft tumor growth in vivo [11], leading to a less aggressive phenotype. We similarly observed a significantly slower tumor growth of the PyMT ITGB1 KO cells. These cells lost the tumor promotion phenotype in response to cardiac remodeling [12]. 

Next, immortalized and primary Mouse Embryonic Fibroblasts (MEF^I^ and MEF^P^, respectively) were used to understand whether the malignancy of the injected cells plays a role. Primary MEF cells play a critical role in induced pluripotent stem cell (iPSC) generation [13]. These cells retain crucial characteristics in culture and secrete various growth factors into the surrounding media [14]. Nonetheless, immortalized MEF cells, engineered to bypass cellular senescence, offer sustained proliferation in culture while retaining primary MEF features [15]. However, due to their tumor-like properties, they may serve as a potential intermediary between cancer cell injection and primary MEF injection in certain contexts.

Remarkably, these cell lines did not lose their ability to improve cardiac dysfunction and reduction in fibrosis in the heart, diaphragm, and skeletal muscles. Similarly, MEF^I^ and MEF^P^ cells displayed the beneficial phenotype in the MDX mouse model; thus, the malignancy and tumor aggressiveness are not required for fibrosis suppression. The effect was mediated at least in part by macrophage polarization in the heart and diaphragm muscles, consistent with our previous finding [8]. These paradigms and the use of even primary MEF cells offer promising opportunities for developing new therapeutic approaches for human fibrotic diseases, which could potentially lead to significant advancements in treatment methods within clinical settings.

## 2. Results

### 2.1. PyMT ITGB1 KO, MEF^I^, and MEF^P^ Cells Ameliorate Heart Contractile Dysfunction and Suppress Fibrosis in the MDX Mice

To examine how PyMT ITGB1 KO, MEF^I^, and MEF^P^ cells affect cardiac dysfunction in the MDX mouse model, cells were injected into the flanks of five-month-old MDX male mice (Figure 1A). Cardiac function was assessed using echocardiography and fractional shortening (FS) was calculated (Figure 1B). While non-injected MDX male mice displayed significantly low contractile function with FS of ~21%, PyMT ITGB1 KO-, MEF^I^-, and MEF^P^-injected mice reached FS of ~27%, 28%, and 31%, respectively. 

Cardiac contractile dysfunction in the MDX mouse model is typically associated with fibrosis in the heart. Therefore, we examined the extent of fibrosis in heart sections derived from control MDX mice compared with PyMT ITGB1 KO-tumor-bearing mice and MEF^I^- and MEF^P^-injected mice by Masson’s Trichrome staining. Indeed, tumor-bearing MDX mice and MEF-injected mice displayed less fibrosis compared to the control MDX cohort (Figure 1C, Appendix A). 

### 2.2. Improved Contractile Function of the Heart Is Accompanied by a Reduction in Fibrosis Hallmark GENE Markers

The improvement in the cardiac function of MDX-injected mice and the reduction in fibrosis staining was accompanied by lower levels of expression of the fibrosis hallmark gene markers, as evaluated by qRT-PCR using mRNA derived from the heart, diaphragm, and skeletal muscles. Collectively, the ITGB1 KO PyMT breast cancer cell line with low aggressive tumor properties was able to recapitulate the beneficial phenotype in the MDX mouse model (Figure 2A, Appendix A). Similarly, MEF^I^- and MEF^P^-injected MDX mice had reduced levels of fibrosis hallmark gene markers compared with non-injected MDX mice, as evaluated by qRT-PCR using mRNA derived from the hearts (Figure 2B,C), diaphragms, and skeletal muscles (Appendix A). 

### 2.3. M2 Macrophage Gene Expression in the Heart

Our previous study showed that tumor growth in MDX mice induced the expression of M2-polarization [5]. Therefore, we sought to examine macrophages’ hallmark gene markers in the heart and diaphragm muscles. Towards this end, we used qRT-PCR analysis of mRNA derived from the hearts and diaphragms of PyMT ITGB1 KO-, MEF^I^- and MEF^P^-injected MDX mice and control MDX mice, respectively. Macrophages are largely divided into two main functionally distinct forms, pro-inflammatory (M1) and anti-inflammatory (M2) [16,17]. To distinguish between the two macrophage populations, we examined the expression levels of specific M1 (*TNFα, IFN-γ, GRP18, IL-1*) and M2 macrophage gene markers (*ARG-1, CCL2, CD163, G-CSF, and IL-13*) [16,17]. We observed higher levels of M2 gene markers and lower levels of M1 gene markers in the heart (Figure 3A–F) of PyMT ITGB1 KO-, MEF^I^- and MEF^P^-injected MDX mice. Of note, a substantial decrease in M1 gene markers in the diaphragm muscles (Appendix A) was observed across all the injected mice cohorts. These results suggest that M1 to M2 polarization occurs in the hearts and most likely the diaphragms of injected MDX mice compared with control MDX mice.

We next examined a master regulatory factor associated with the M1 to M2 macrophage switch, IL-13 [18], in the serum derived from control MDX mice and PyMT ITGB1- KO-, MEF^I^- and MEF^P^-injected MDX mice. Significantly, higher levels of IL-13 were observed in the serum of PyMT ITGB1- KO-, MEF^I^- and MEF^P^-injected MDX mice as compared with control MDX mice (Figure 4). This observation is consistent with the observation of M2 macrophages polarization in the fibrotic muscle tissues of MDX mice.

Together, our results suggest that MDX mice injected with various cell lines exhibit improved cardiac contractile dysfunction with lower muscle fibrosis. Remarkably, the M2 macrophages switch occurs in the fibrotic muscles. A schematic summary of the major findings and conclusions of the manuscript is depicted in Figure 5.

## 3. Discussion

Fibrosis represents an unresolved clinical challenge in numerous diseases [19]. One such disease is DMD, an X-linked genetic disease, that involves fibrosis in cardiac, diaphragm, and skeletal muscles [20]. While recent studies have shown that injecting two aggressive murine cancers (PyMT and LLC cell line) reduces cardiac hypertrophy and fibrosis and enhances cardiac function in multiple mouse models [2,3,5], it was not addressed whether the aggressiveness and nature of cancer types impacted this improvement. Toward this end, we employed the MDX mouse model, characterized by fibrosis and cardiac dysfunction [20,21], to evaluate heart contractile function. Following the injection of three types of cells: less aggressive PyMT cells (*ITGB1* KO), immortalized MEF cells (MEF^I^), and primary MEF cells (MEF^P^), all cell lines in MDX mice improved the cardiac contractile function. Moreover, there was a significant reduction in the fibrosis staining and suppressed mRNA levels of hallmark gene markers transcription in the heart, diaphragm, and skeletal muscles of MDX mice. Previously, we have shown that the amelioration of cardiac dysfunction and the suppression of fibrosis occurs in a gender-independent manner [5]. 

These findings strongly indicate that the injection of these three cell lines inhibits the formation of new fibrosis, as evidenced by the modulation of hallmark gene markers. This observation aligns closely with our earlier findings in tumor-bearing MDX mice [5]. Moreover, the reduced fibrosis hallmark gene markers expression occurs in all muscle tissues. Immortalized and primary MEF cell lines are known to secrete various growth factors that can modulate cellular responses and tissue microenvironments [15,22]. Notably, primary MEF cell line are called “feeder cells” and are used to maintain the pluripotent state of stem cells [14]. The secretion of these growth factors by MEF cells could stimulate anti-fibrotic processes, such as the regulation of cell proliferation, migration, and differentiation when injected into the MDX mice. Additionally, MEF cells have the capacity to undergo differentiation into diverse cell types under appropriate conditions [23]. Therefore, when injected into the tissue, MEF cells may differentiate into cell types that contribute to the suppression of fibrosis, leading to an overall improvement in tissue condition. Previously, cell therapy interventions for DMD primarily focused on stem cell therapy [24,25,26]. Recently, mitochondrial transplantation has emerged as a promising approach [27], adding a new dimension to DMD treatment options. In our ongoing research, we are investigating factors secreted by cancerous and non-cancerous cell lines to understand their roles in reducing fibrosis and to assess their therapeutic potential. Cells secrete numerous factors that have the potential to influence the pathology of fibrotic disease. For instance, cancer cells may secrete antioxidant enzymes that may mitigate oxidative stress in Duchenne dystrophy [28]; additionally, cells may absorb calcium ions, crucial in this fibrosis pathology. Unrevealing these mechanisms may lead to identification of novel therapy strategies for Duchenne dystrophy treatment and other fibrotic diseases.

Overall, the injection of these individual cell lines, despite their different characteristics, can lead to similar improvements in fibrosis due to their ability to modulate cellular responses, secrete growth factors, and potentially undergo differentiation, all of which contribute to the mitigation of fibrotic and inflammatory processes in the tissue.

This was validated through qRT-PCR and ELISA utilizing M2-polarizing secreted factors in the serum. Particularly in the heart, there was an increase in the local production of well-known and established M2-polarizing cytokines (GCSF, IL13, CCL2, ARG1, and CD163) [29]. Notably, CCL2 has been demonstrated to have a significant role in mitigating left ventricular dysfunction and remodeling post-myocardial infarction (MI) [30]. M2 macrophages are hypothesized to expedite the progression of cardiac and tissue repair mechanisms [16,31]. In contrast, we observed a parallel decrease in pro-inflammatory M1-polarizing cytokines IL1, TNFα, INFγ, and GPR18 [29,32] in the heart and diaphragm muscle. The definitive polarization of macrophages occurs upon their localization at the fibrotic site, notably observed in organs such as the heart and diaphragm muscles [33]. Moreover, ELISA demonstrated increased levels of the M2-polarizing factor IL-13 in these mice compared to control MDX mice.

Significantly, the involvement of M2 macrophages in suppressing fibrosis across diverse organs underscores the clinical importance of comprehending the mechanisms underlying macrophage polarization following injection with these cell lines. A previous study showed that IL-13 induces M2 polarization, leading to improved cardiac function and reduced heart injury in a viral myocarditis mouse model [19]. It was also shown that the M2 population supports fibrosis, whereas the increase in the M2 population in tumor-bearing MDX mice appears to be associated with the repair and improvement in cardiac function [5]. In the current paper, injection of ITGB KO and MEFs improved cardiac function at least partially due to macrophage polarization, which was shown to be mediated via IL-13 regulation. This approach may also have a beneficial systemic outcome for other fibrotic diseases. 

Limitations: In DMD human patients, fibrosis treatment is limited, and cancer is not considered an optional treatment. The identification of factors that are responsible for the beneficial effects here may be used as a potential treatment. Additionally, we lack understanding regarding the viability of the cells injected into the MDX mice. This could be significant for their secretion of factors that may confer therapeutic effects in this model. In addition, while qRT-PCR is provided for cardiac, skeletal, and diaphragm muscles, histological data are provided for cardiac muscles only. Further histological assessments may be necessary to extend the beneficial effects to other fibrotic muscles beyond the mRNA levels.

Clinical perspectives: The manuscript describes a beneficial effect of less aggressive cancer and non-cancerous approach on cardiac dysfunction and fibrosis in a clinically relevant mouse model. Fibrosis diseases account for more than half of deaths worldwide and represent an unmet need. Utilizing this paradigm could offer a new approach to enhancing cardiac function and addressing fibrosis-related disorders.

## 4. Materials and Methods

### 4.1. Animals

The C57BL/10ScSn-Dmd (MDX) mice were purchased from the Jackson Laboratory. The mice were bred and nurtured at the Pre-Clinical Research Authority, Ruth and Bruce Rappaport Faculty of Medicine, Technion. Upon reaching three weeks of age, pups were weaned to individual cages. 

### 4.2. Cell Lines 

PyMT ITGB1 KO cells were generated as previously described [12]. Immortalized Mouse embryo fibroblast (MEF^I^) cell lines [34] and primary Mouse embryo fibroblast (MEF^P^) were generated as previously described [35]. Cells were cultured in DMEM containing 10% FBS, 1% streptomycin and penicillin, 1% l-glutamine, and 1% sodium pyruvate at 37 °C in a humidified atmosphere containing 5% CO_2_. MEF cell implantation was conducted at maximal passage number five (MEF^I^) and passage 2 (MEF^P^). 

### 4.3. Cell Implantation

PyMT ITGB1 KO, MEF^I^, and MEF^P^ were injected (10^6^ cells per mouse) into the back flanks of mice. 

### 4.4. Echocardiography 

The mice were anesthetized with 1% isoflurane and kept on a 37 °C heated plate throughout the procedure. Echocardiography was performed with a Vevo3100 micro-ultrasound imaging system (VisualSonics, Fujifilm, Tokyo, Japan), equipped with 13–38 MHz (MS 400) and 22–55 MHz (MS550D) linear array transducers. Cardiac size, shape, and function were analyzed using conventional two-dimensional imaging and M-mode recordings. Maximal left ventricular end-diastolic (LVDd) and end-systolic (LVDs) dimensions were measured in short-axis M-mode images. Fractional shortening (FS) was calculated with the following formula: FS% = [(LVDd − LVDs)/LVIDd] × 100. FS values were based on the average of at least three measurements for each mouse. 

### 4.5. RNA Extraction 

RNA was extracted from hearts, diaphragms, and skeletal muscle (Tibialis Anterior) using an Aurum total RNA fatty or fibrous tissue kit (no. 732–6830, Bio-Rad, Hercules, CA, USA), according to the manufacturer’s instructions. cDNA was synthesized from 1000 ng purified mRNA with an iScript cDNA Synthesis Kit (no. 170–8891, Bio-Rad), according to the manufacturer’s instructions. 

### 4.6. Quantitative Real-Time PCR 

Quantitative real-time polymerase chain reaction (qRT-PCR) was performed with a QuantStudio3 (Thermofisher Scientific, 5823 Newton drive, Carlsbad, CA, USA). Serial dilutions of a standard sample were included for each gene to generate a standard curve. Values were normalized to mb2M expression levels for the heart and diaphragm muscles. All the oligonucleotide sequences that were used are listed in Appendix A. 

### 4.7. Fibrosis Staining 

Heart tissues and diaphragms were fixed in 4% formaldehyde overnight, embedded in paraffin, serially sectioned at 10 μm intervals, and then mounted on slides. Masson trichrome staining was performed according to the standard protocol. Images were ac-quired using a 3DHistech Pannoramic 250 Flash III (3DHISTECH Ltd., H-1141 Budapest, Öv u. 3., Hungary). Each section was fully scanned. The percent of interstitial fibrosis was determined as the ratio of the fibrosis area to the total area of the section using using Image Pro Priemier 9.3.3 software. Each dot represents the mean of the values taken from at least five fields, derived from a single mouse. 

### 4.8. ELISA 

ELISA quantification of the Mouse IL-13 ELISA Kit (M1300CB, R&D systems Inc., Minneapolis, MN, USA) was performed according to the manufacturer’s instructions. 

### 4.9. Statistical Analysis

The data are presented as mean ± standard error (SE). All mice were included in each statistical analysis, unless they were euthanized for humane reasons before reaching the experimental endpoint. During data collection, the experimental groups were blinded to the researchers. The mice for each group were selected randomly. Each experimental group consisted of at least *n* = 4 mice. To determine the statistical significance of tumor volume, a two-way repeated-measures ANOVA followed by the Bonferroni post-test was used. For comparisons between several means, a one-way ANOVA followed by the Tukey post-test was performed. For comparisons between two means, either a two-tailed Student’s *t*-test or Mann–Whitney U test was utilized. All statistical analyses were conducted using GraphPad Prism 10 software. A significance level of *p* < 0.05 was considered statistically significant. 

## Figures and Tables

**Figure 1 ijms-25-04273-f001:**
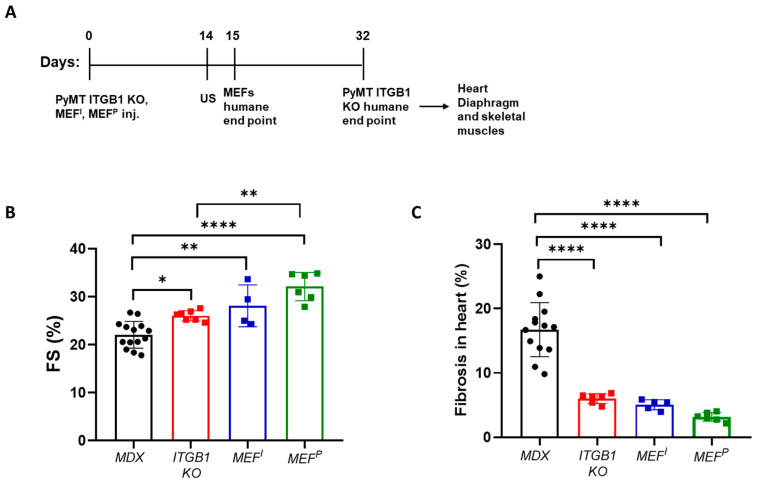
Injection of less aggressive PyMT ITGB1 KO cells and non-cancerous MEF^I^ and MEF^P^ cells resulted in a significant improvement of cardiac contractile function in MDX mice. (**A**) Schematic representation of the experimental timeline. MDX male mice (5 months old) were injected in the flanks with PyMT ITGB1 KO (red), MEF^P^ (blue), or MEF^P^ (green) cells (10^6^ cells per mouse) or left non-injected (MDX) (black). Echocardiography (US) was performed 14 days after injection. (**B**) The measured fractional shortening (FS) of control MDX and injected MDX mice group (ITGB1 KO, MEF^P^, and MEF^I^). FS was assessed by echocardiography and calculated using the formula: FS (%) = [(LVDd − LVDs)/LVDd]. (**C**) Percent of interstitial fibrosis of all injected groups of MDX mice compared with control MDX mice, quantified using ImageJ 1.53t software, based on at least five fields from each mouse in each cohort. Data are presented as mean ± SE. One-way ANOVA followed by Tukey post-test (**B**,**C**). * *p* < 0.05; ** *p* < 0.01; **** *p* < 0.0001. Each dot represents one mouse.

**Figure 2 ijms-25-04273-f002:**
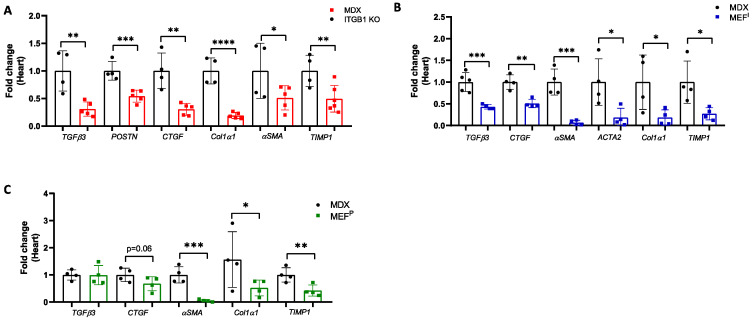
Injection of less aggressive PyMT ITGB1 KO cells and non-cancerous MEF^I^ and MEF^P^ cells led to a reduction in the expression of fibrosis hallmark gene markers in the hearts of MDX mice. (**A**–**C**) qRT-PCR measuring transcription mRNA levels of fibrosis hallmark gene markers in the hearts of MDX mice injected with (**A**) PyMT ITGB1 KO, (**B**) MEF^I^, and (**C**) MEF^P^, as compared with control MDX. Data are presented as mean ± SE. One-way ANOVA followed by Tukey post-test (**A**–**C**). * *p* < 0.05; ** *p* < 0.01; *** *p* < 0.001; **** *p* < 0.0001. Each dot represents one mouse.

**Figure 3 ijms-25-04273-f003:**
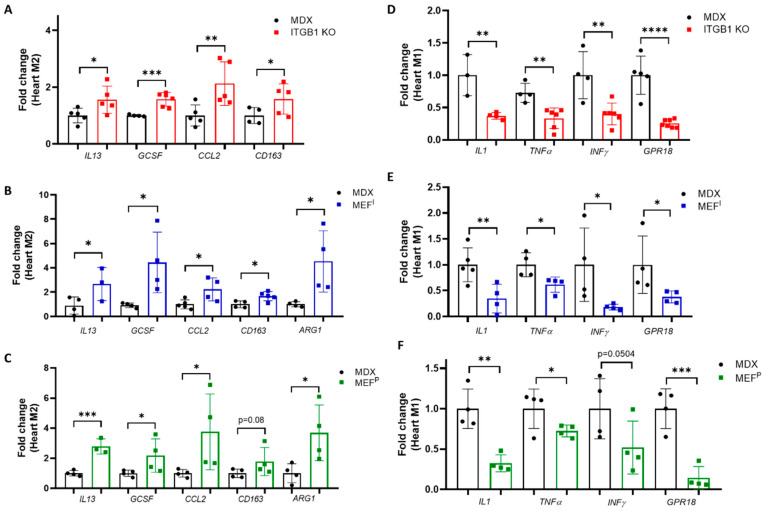
Injection of less aggressive cancer cell line PyMT ITGB1 KO cells and non-cancerous MEF^I^ and MEF^P^ cells induces the expression of M2-polarizing hallmark gene markers in the hearts of MDX mice. (**A**–**F**) qRT-PCR measures the transcription levels of macrophage hallmark gene markers in the hearts of injected MDX mice as compared with control MDX (**A**–**C**) mRNA levels of different M2 macrophage markers: *IL13, GCSF, CCL2, CD163, and ARG1*, (**D**–**F**) mRNA levels of different M1 macrophage markers: *IL1, TNFα, INFγ and GPR18*. Measurements were obtained using qRT-PCR, normalized to the housekeeping gene mB2M. The results are presented as mean ± SE, one-way ANOVA followed by Tukey post-test * *p* < 0.05; ** *p* < 0.01; *** *p* < 0.001; **** *p* < 0.0001. Each dot represents one mouse.

**Figure 4 ijms-25-04273-f004:**
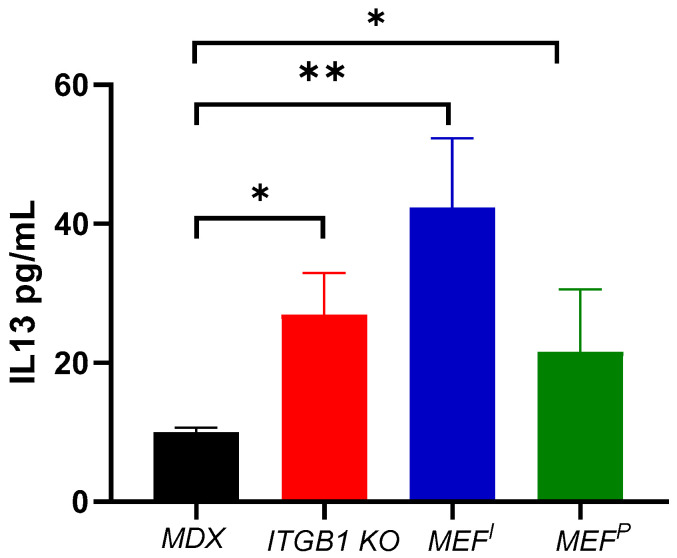
M2 polarizing cytokines IL-13 is elevated in the serum of injected mice (PyMT ITGB1 KO, MEF^I^, and MEF^P^) compared with control MDX mice. Serum levels are obtained by ELISA for IL-13. Pooled blood sera of control MDX (*n* = 3 each), tumor-bearing (ITGB1 KO) MDX mice (*n* = 4), MEF^I^-injected MDX mice (*n* = 3) and MEF^P^-injected MDX mice (*n* = 4) were used. The results are presented as mean ± SE. One-way ANOVA followed by Tukey post-test. * *p* < 0.05, ** *p* < 0.01.

**Figure 5 ijms-25-04273-f005:**
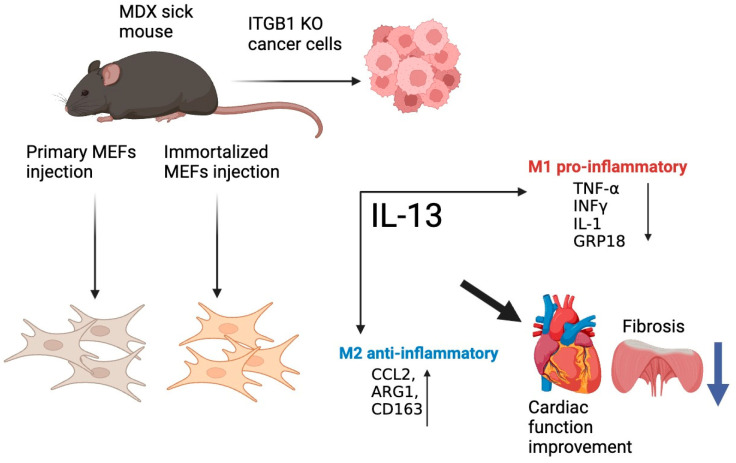
Graphical abstract summarizing the conclusions of this study. Tumor progression is not necessary for the beneficial phenotype and reduced fibrosis in the cardiac, skeletal, and diaphragm muscles. These effects are attributed, at least in part, to the M2 macrophage switch.

## Data Availability

All the obtained data used to support the findings of this study are available from the corresponding author upon request.

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
