# Peer review of "Cardioprotection and Suppression of Fibrosis by Diverse Cancer and Non-Cancer Cell Lines in a Murine Model of Duchenne Muscular Dystrophy"

_ijms, 2024, doi:10.3390/ijms25084273_

Round 1

Reviewer 1 Report

Comments and Suggestions for Authors

The manuscript, ' Cardioprotection and suppression of fibrosis by diverse cancer and non-cancer cell lines in a murine model of Duchenne muscular dystrophy,’ is very interesting and well-written. The authors demonstrated the role of two different cell types in ameliorating cardiac dysfunction in MDX mice and the role of these two cells in improving the fibrotic phenotype in the diaphragm and skeletal muscles. I would appreciate it if the authors addressed the following concerns.

 1.        This study does not have any echo/ultrasound data for the baseline (just before the implantation of cells) to assess the cardiac parameters (e.g., FS%) in the same mice. It is very important to compare pre-and post-implantation ultrasound data using the same mice. Are there any changes in the EF (%) and/or RWT?

 2.        Are there any changes in the HW/BW or LV/BW in the mice?

 3.        How long do the MDX mice survive? Adding a survival curve would help to see the effects of cell (PyMT or MEF) implantation on reduced mortality due to amelioration of DMD.

Reviewer 2 Report

Comments and Suggestions for Authors

Interesting work with well-structured presentation of results. The statistics are at a high level, laying the foundation for future prospects of introducing new approaches to the treatment of cardiac dysfunction. This work can be accepted for publication.

Reviewer 3 Report

Comments and Suggestions for Authors

In a series of recent papers, the authors show that administration of diverse cancer and non-cancer cell lines to dystrophin-deficient mdx mice has a therapeutic effect, improving the state of both cardiac and skeletal muscles. It has been suggested that the introduction of cells may be accompanied by an increase in the polarization of M2 macrophages and suppression of fibrosis. The authors obtained very interesting results. However, in its present form, the work requires a series of additional experiments:

1. First of all, the authors should provide the necessary control groups, these should be C57BL/10 mice, which will be injected with the vehicle and the corresponding cell lines, this will eliminate some nonspecific effects of the tested therapy. This is a critical remark. Also, the work does not indicate what was administered to the control mdx animals. I think that this should be some kind of medium. The authors should also describe what medium was used to resuspend the cells and administer them to the animals. It is important that the influence of components of the incubation medium should be excluded.

2. If the authors focus on cardiac pathology, then it is critical for them to consider the effect of therapy on female mdx mice, which exhibit a more pronounced cardiac phenotype, in contrast to males (and 5 months is short enough for the development of significant cardiac pathology in male mdx mice).

3. What happens to the cells after they are introduced? Do they maintain their viability? This must be important for their secretion of factors that, according to the authors, have therapeutic effects.

4. Justify the method of introducing cells, as well as the chosen concentration. Why didn't the authors normalize the number of cells to the weight of the animals?

5. The supplementary materials indicate that the state of the diaphragm and skeletal muscles was also assessed (it is not indicated which skeletal muscles were used for analysis, this is critical). The Materials and Methods do not indicate what was analyzed in these muscles. I also recommend that the authors evaluate the level of centrally nucleated fibers in skeletal muscles; this is a more indicative criterion, since fibrosis develops poorly in the mdx mouse line.

6. Statistical analysis requires combining the results of experiments using different cell lines and comparing their performance using ANOVA.

7. The explanation is too one-sided, although the authors have some remarkable results. Cell therapy has already been used to treat Duchenne dystrophy in mdx mice. Cells contain many factors that can affect the development of pathology; moreover, even the immune response and the development of stress can have a positive effect on the body. Cells contain many enzymes, for example, cancer cells are rich in antioxidant enzymes, which may help scavenge reactive oxygen species and alleviate the oxidative stress associated with Duchenne dystrophy. Moreover, they can absorb calcium ions, which is important in the case of this pathology. In particular, even the introduction of intracellular organelles (for example, mitochondria, as recently shown) reduces calcium overload in muscle fibers and mitigates the development of Duchenne dystrophy. This cannot be excluded in the case of the approach proposed by the authors. It may also contribute to the trafficking of cell adhesion complexes to the sarcolemma and mitigate the development of pathology, as shown in the case of mdx and δ-sarcoglycan-deficient mice.

Round 2

Reviewer 1 Report

Comments and Suggestions for Authors

Thanks to the authors for addressing my concerns with the previous version of the manuscript!

Author Response

Thanks to the authors for addressing my concerns with the previous version of the manuscript!

Response: We wish to thank the reviewer who found our revised version of our manuscript sufficient and suitable for publication.

Reviewer 3 Report

Comments and Suggestions for Authors

Unfortunately, the authors did not react in any way to my comments and remarks, although they are very critical.

1. To my main criticism, the authors respond that «In our previous study, as detailed in Achlaug et al. (reference #5), we investigated the impact of injecting cancer cells, specifically the PyMT and LLC cell lines, on the cardiac function of MDX in comparison to control C57BL/10 mice. Our results revealed that for C57Bl10 there were no effects on cardiac function, as indicated by fractional shortening (FS%) remaining consistent before and after injection. Interestingly, our findings suggest that any observed effects of cancer cell injection are likely mediated on cardiac dysfunction related to fibrosis».

However, in the presented article, the authors introduced completely different cell lines - PyMT breast cancer cell line lacking integrin β1, as well as immortalized and primary MEF cells. Can the authors ensure that these cell lines have the same characteristics as the original PyMT and LLC cell lines, secrete the same factors, and have the same effect on the phenotype of mdx and wild-type mice? If this is the case, then the authors need to provide this data. If there is no such data, the authors are obliged to conduct the indicated experiments on control animals. As I wrote, this is a mandatory point.

2. The authors write that «In our previous publication by Achlaug et al. (reference #5) we conducted experiments involving male and female MDX mice. Notably, our analysis revealed no significant differences in cardiac function between male and female MDX mice (roughly in the same age group). This observation was evident as fibrosis developed almost concurrently in both genders, and the effects of cancer were similarly significant in both male and female mice.

The authors should discuss these results in the context of the new data.

3. Regarding my comment « What happens to the cells after they are introduced? Do they maintain their viability? This must be important for their secretion of factors that, according to the authors, have therapeutic effects».

I understand that not everything can be taken into account when conducting such large-scale studies. However, the authors should discuss this point as a possible limitation of the study.

4. Regarding the need to evaluate the histological parameters of the diaphragm and skeletal muscles (previous comment 5). Indeed, they provide data on changes in the level of gene expression in these muscles. However, this does not necessarily lead to changes at the tissue level. The authors provide histological data on cardiac muscle, which are in good agreement with data on gene expression in this tissue. However, the effect may be tissue specific. Therefore, the authors need to provide similar histological data for other muscles (diaphragm and Tibialis Anterior) in order to relate these data to the level of gene expression. Alternatively, the authors could exclude these data and focus only on cardiac muscle.

5. Regarding statistical data processing. I meant that the authors needed to conduct a simultaneous statistical analysis of all groups used in the work using ANOVA. This will reveal differences in effects between all groups.

6. Regarding my criticism of the discussion of the results. I understand that the authors are at the beginning of a long journey and some studies cannot be completed simultaneously at this stage. However, this does not prevent them from discussing all the possible options that I have indicated and those mechanisms that I may not have mentioned. This will set a new vector for research and indicate possible problems that need to be addressed in future experiments.  

Round 3

Reviewer 3 Report

Comments and Suggestions for Authors

The authors partially answered my questions, although the answer to the first question is discouraging. The scientific approach requires confirmation of results (and not expectation), no matter how great the probability of obtaining certain results may seem.

A couple of comments:

1. The authors write that «The lack of IHC staining for these tissues will be added to the limitation statement.».

I didn't find any mention of this in the text.

2. Regarding statistical data processing.

The authors provided comparative data on the cardiac function of MDX mice before and after injection of different cell lines in the file with the authors’ response. However, nothing has changed in the main text of the manuscript. The authors should process and analyze (importantly) in this manner the data presented in Figures 1 B,D + 2B, D + 3 B,D. Only Figure 5 is shown correctly (all groups are compared with each other).

Round 4

Reviewer 3 Report

Comments and Suggestions for Authors

The authors responded to the latest comments